# CDK4/6 inhibition mitigates stem cell damage in a novel model for taxane-induced alopecia

Talveen S Purba[1],[*] iD, Kayumba Ng'andu[1], Lars Brunken[2], Eleanor Smart[1] iD, Ellen Mitchell[1], Nashat Hassan[1], Aaron O'Brien[1], Charlotte Mellor[1], Jennifer Jackson[1], Asim Shahmalak[3] & Ralf Paus[1,2,4,**] iD

## Abstract

**Taxanes are a leading cause of severe and often permanent chemotherapy-induced alopecia. As the underlying pathobiology of taxane chemotherapy-induced alopecia remains poorly understood, we investigated how paclitaxel and docetaxel damage human scalp hair follicles in a clinically relevant *ex vivo* organ culture model. Paclitaxel and docetaxel induced massive mitotic defects and apoptosis in transit amplifying hair matrix keratinocytes and within epithelial stem/progenitor cell-rich outer root sheath compartments, including within Keratin 15+ cell populations, thus implicating direct damage to stem/progenitor cells as an explanation for the severity and permanence of taxane chemotherapy-induced alopecia. Moreover, by administering the CDK4/6 inhibitor palbociclib, we show that transit amplifying and stem/progenitor cells can be protected from paclitaxel cytotoxicity through G1 arrest, without premature catagen induction and additional hair follicle damage. Thus, the current study elucidates the pathobiology of taxane chemotherapy-induced alopecia, highlights the paramount importance of epithelial stem/progenitor cell-protective therapy in taxane-based oncotherapy, and provides preclinical proof-of-principle in a healthy human (mini-) organ that G1 arrest therapy can limit taxane-induced tissue damage.**

**Keywords** chemotherapy; hair loss; palbociclib; taxol; taxotere
**Subject Categories** Cancer; Pharmacology & Drug Discovery

## Introduction

Chemotherapy-induced alopecia is a highly distressing adverse effect of cancer treatment and can persist long after the completion of chemotherapy treatment regimens (Paus *et al*, 2013). As many as 8% of patients have been found to be at risk of rejecting chemotherapy due to the psychosocial burden imposed by chemotherapy-induced alopecia, which is detrimental to patient self-esteem, body image and quality of life (McGarvey *et al*, 2001) especially when the effects of chemotherapy are permanent (Freites-Martinez *et al*, 2019). The only currently available preventive treatment for chemotherapy-induced alopecia is scalp cooling, whose clinical efficacy is as yet unsatisfactory and difficult to predict, especially with taxane chemotherapy-induced alopecia (Friedrichs & Carstensen, 2014; Cigler *et al*, 2015; Rugo *et al*, 2017; Rice *et al*, 2018). Furthermore, scalp cooling does not extend protection against hair loss to other body sites of cosmetic, cultural, religious and psychosocial relevance, e.g. eyebrow, beard or pubic hair.

Therefore, novel and effective chemotherapy-induced alopecia prevention strategies need to be urgently developed and translated into clinical practice. This can only be achieved through the generation of promising preclinical data in appropriate human models that are as close as possible to clinical chemotherapy-induced alopecia (Bodó *et al*, 2007, 2009; Paus *et al*, 2013; Böhm *et al*, 2014; Sharova *et al*, 2014; Yoon *et al*, 2016).

To date, preclinical chemotherapy-induced alopecia research models have been developed to study how doxorubicin and cyclophosphamide damage the human hair follicle (Bodó *et al*, 2007, 2009; Paus *et al*, 2013; Böhm *et al*, 2014; Sharova *et al*, 2014; Yoon *et al*, 2016). However, the field currently lacks a model of how taxanes, major current oncotherapeutics used to treat breast and lung cancer, damage the human hair follicle and cause chemotherapy-induced alopecia. The need for such a model is becoming increasingly important, given the abundance of reports describing permanent taxane chemotherapy-induced alopecia (Prevezas *et al*, 2009; Tallon *et al*, 2010; Miteva *et al*, 2011; Palamaras *et al*, 2011; Kluger *et al*, 2012; Tosti *et al*, 2013; Sibaud *et al*, 2016; Kang *et al*, 2018; Martín *et al*, 2018). This is often reported following treatment with docetaxel, which is the subject matter of on-going lawsuits against Taxotere (docetaxel) manufacturer Sanofi (Raymond, 2019).

Taxanes are microtubule-stabilising agents whose principle antineoplastic mode of action is through the disruption of mitosis, e.g. by promoting chromosome missegregation/cell division on multipolar spindles (Chen & Horwitz, 2002; Abal *et al*, 2003; Morse *et al*,

1 Centre for Dermatology Research, School of Biological Sciences, University of Manchester & NIHR Biomedical Research Centre, Manchester, UK
2 Monasterium Laboratory – Skin & Hair Research Solutions GmbH, Münster, Germany
3 Crown Clinic, Manchester, UK
4 Dr. Phillip Frost Department of Dermatology & Cutaneous Surgery, University of Miami Miller School of Medicine, Miami, FL, USA
 *Corresponding author. Tel: +44 161 275 5382; E-mail: talveen.purba@manchester.ac.uk
 **Corresponding author. Tel: +1 305 243 7870; E-mails: rxp803@med.miami.edu; ralf.paus@manchester.ac.uk

2005; Weaver, 2014; Zasadil *et al*, 2014). Taxanes are thus presumed to cause hair loss by damaging rapidly dividing matrix keratinocytes and their counterpart stem/progenitor cells, required for healthy hair growth and hair follicle cycling (Paus & Cotsarelis, 1999; Garza *et al*, 2011; Paus *et al*, 2013; Purba *et al*, 2016, 2017a,b; Gao *et al*, 2019; Huang *et al*, 2019). However, the effects of taxane chemotherapy on the human hair follicle remain to be systematically examined.

Therefore, in the current study we aimed to develop a clinically relevant *ex vivo* assay for studying and experimentally manipulating taxane toxicology in healthy human hair follicles to elucidate how taxanes cause chemotherapy-induced alopecia. To do so, we used a well-established *ex vivo* organ culture model (Langan *et al*, 2015) to dissect how the taxanes paclitaxel and docetaxel damage full-length human anagen VI scalp hair follicles. Specifically, we focused on how the mitosis-targeting cytotoxicity of taxanes affected highly proliferative hair-forming matrix keratinocytes (Purba *et al*, 2016, 2017a). Furthermore, we also asked whether taxanes damage (relatively slow-cycling) epithelial stem/progenitor cell niches in the hair follicle outer root sheath (Garza *et al*, 2011; Purba *et al*, 2017b), especially as irreversible stem/progenitor cell damage may lead to permanent chemotherapy-induced alopecia (Paus *et al*, 2013).

Given that the mode of action of taxanes relies upon direct interference with the cell cycle (i.e. mitosis) to initiate tumour cell death (Chen & Horwitz, 2002; Abal *et al*, 2003; Morse *et al*, 2005; Weaver, 2014; Zasadil *et al*, 2014), we further probed in our newly developed taxane chemotherapy-induced alopecia model the working hypothesis that pharmacologically induced cell cycle arrest protects against taxane-induced human hair follicle damage (Shah & Schwartz, 2001; Blagosklonny, 2011; McClendon *et al*, 2012; Paus *et al*, 2013; Beaumont *et al*, 2016). To achieve this, we used the G1 arresting CDK4/6 inhibitor palbociclib, which is used in the treatment of hormone (oestrogen and/or progesterone) receptor-positive HER2-negative breast cancer (Ro *et al*, 2015). Palbociclib was employed as previous reports have described that pharmacological CDK4/6 inhibition can protect against chemotherapy-induced acute kidney injury (DiRocco *et al*, 2014; Pabla *et al*, 2015) and chemotherapy-induced haematopoietic stem cell exhaustion (He *et al*, 2017).

With this experimental approach, we have probed whether palbociclib is a candidate chemotherapy-induced alopecia-preventive lead agent and sought to obtain a proof-of-principle that pharmacologically induced cell cycle arrest can protect transit amplifying matrix keratinocytes and epithelial stem/progenitor cells within their native tissue habitat from chemotherapy-induced apoptosis, ideally without promoting premature hair follicle regression (catagen) (Paus *et al*, 2013).

# Results

## Taxanes induce the massive accumulation of phospho-histone H3+ cells in the anagen matrix of human scalp hair follicles

To determine the effects of taxane chemotherapy on the most rapidly proliferating keratinocytes of human scalp hair follicles, i.e. anagen hair matrix keratinocytes (Purba *et al*, 2016, 2017a), we first treated microdissected, organ-cultured human hair follicles (Langan *et al*, 2015) with 100 nM paclitaxel for 24 h, i.e. at a dose that resembles reported plasma concentrations of paclitaxel 20 h post-infusion (Zasadil *et al*, 2014).

*In situ* cell cycle analyses (Purba *et al*, 2016) revealed that paclitaxel exerts mitosis-specific effects on proliferating human hair follicle matrix keratinocytes, rather than globally inhibiting proliferation. In fact, as a marker of global cell cycle activity, an analysis of the total number of Ki-67+ cells in the hair matrix showed no statistically significant difference between vehicle- and paclitaxel-treated hair follicles (Fig 1A). Moreover, EdU incorporation within the hair matrix *ex vivo* revealed no significant effect on the number of cells in S-phase (i.e. undergoing DNA synthesis) following 24-h paclitaxel treatment (Fig 1B).

However, paclitaxel promoted a large and significant increase in the number of cells labelled positively with the mitosis-specific marker phospho-histone H3 (pH3) (Crosio *et al*, 2009; Purba *et al*, 2016; Fig 1C). Treatment with 100 nM docetaxel for 24 h also induced a profound increase in the number of pH3+ cells in the anagen hair matrix, confirming this effect on the human hair follicle as a shared feature of taxanes (Fig 1D). These results reveal that taxanes promote the abnormal accumulation of mitotic (i.e. pH3+) keratinocytes in the hair matrix, signifying mitotic arrest, without affecting G1/S cell cycle progression (Fig 1Ei–iii). Together, these observations are consistent with the recognized mitosis-specific cytotoxicity of taxanes (Jordan *et al*, 1996; Chen & Horwitz, 2002; Abal *et al*, 2003; Morse *et al*, 2005; Weaver, 2014; Zasadil *et al*, 2014; Mitchison *et al*, 2017) and validate the usefulness of our *ex vivo* model for studying taxane toxicity in a rapidly proliferating, healthy human mini-organ.

## Taxanes promote micronucleation, transcriptional arrest and apoptosis in hair matrix keratinocytes

To examine the nuclear morphology of matrix keratinocytes following 24-h paclitaxel and docetaxel treatment, we stained nuclei with Hoechst 33342. Paclitaxel promoted the extensive accumulation of irregular and shrunken nuclei that localised specifically to the most proliferative region of the hair matrix (Fig 2A; i.e. predominantly below the critical line of Auber; Purba *et al*, 2016, 2017a). Docetaxel treatment (24 h) also promoted the formation of irregular nuclear bodies within the hair matrix, albeit not to the extent seen following paclitaxel treatment (Fig 2B–E). These nuclear abnormalities are likely a consequence of mitosis defects, i.e. chromosome missegregation (Chen & Horwitz, 2002; Abal *et al*, 2003; Morse *et al*, 2005; Weaver, 2014; Zasadil *et al*, 2014), giving rise to micronucleated cells. This well-defined taxane-induced phenomenon (Morse *et al*, 2005; Mitchison *et al*, 2017) is a hallmark of "mitotic catastrophe", whereby failed mitosis ultimately leads to cell death or senescence (Vakifahmetoglu *et al*, 2008; Vitale *et al*, 2011).

Next, as a toxicological read-out parameter, we analysed how paclitaxel treatment affects *in situ* global RNA synthesis in the hair matrix through the detection of ethynyl uridine (EU) incorporated during *ex vivo* human hair follicle organ culture, using the recently described methodology (Purba *et al*, 2018). The incorporation of EU was found to be significantly decreased within keratinocytes of the proliferative hair matrix (Fig 3A and B). However, this did not represent a generalised, hair follicle-wide systemic RNA synthesis inhibitory effect (e.g. as seen following broad spectrum CDK inhibition in the human hair follicle; Purba *et al*, 2018). Instead, the

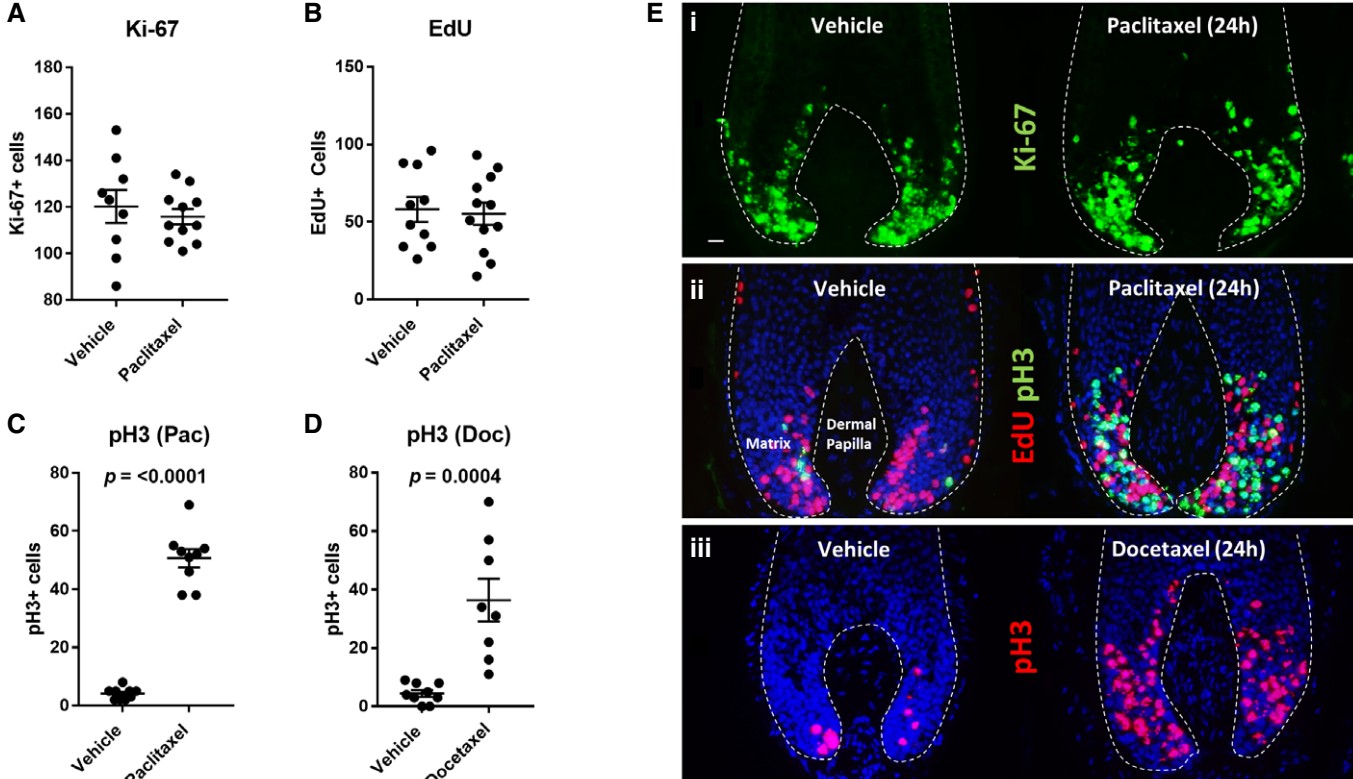

**Figure 1. Taxanes increase the number of phospho-histone H3$^+$ cells in the human anagen hair follicle matrix.**

A, B  100 nM paclitaxel treatment of human hair follicles (HFs) in organ culture for 24 h does not significantly affect the total number of Ki-67$^+$ cells (A) and EdU$^+$ cells (B) (S-phase) in the hair matrix. Unpaired $t$-test performed using $N$ of 9–12 HFs from three patients.

C  100 nM paclitaxel treatment (24 h) significantly ($P \leq 0.0001$) increases the number of mitotic phospho-histone H3 (pH3)$^+$ cells in the hair matrix. Welch's $t$-test performed using $N$ of nine HFs from three patients.

D  100 nM docetaxel treatment (24 h) significantly ($P = 0.0004$) increases the number of pH3$^+$ cells in the hair matrix. Unpaired $t$-test performed using $N$ of 8–9 HFs from three patients.

E  Representative immunofluorescence images highlight the effects of 24-h 100 nM taxane treatment on (i) Ki-67 expression [paclitaxel]; (ii) EdU incorporation and pH3 immunoreactivity [paclitaxel]; (iii) pH3 immunoreactivity [docetaxel]. 20-μm scale.

Data information: Error bars are standard error of the mean. Values plotted represent the mean number of positive cells counted per HF analysed.
Source data are available online for this figure.

population of cells in the hair matrix that failed to incorporate EU was mainly restricted to cells demarcated by pH3 immunoreactivity (Fig 3C). This demonstrates that RNA transcription is attenuated in abnormally dividing/arrested hair matrix keratinocytes following paclitaxel treatment. This could contribute towards the cytotoxicity of taxanes in the hair follicle, as transcriptional arrest during abnormal mitosis may promote cell death (Blagosklonny, 2007).

Paclitaxel treatment significantly increased the number of apoptotic (cleaved caspase-3$^+$) cells within the proliferative hair matrix following 24-h treatment (Fig 3D and E). However, 24-h paclitaxel treatment did not immediately increase cleaved caspase-3 immunoreactivity in hair follicles from all donors, in contrast to the consistent accumulation of pH3$^+$ cells in all hair follicles treated with paclitaxel for 24 h, irrespective of donor (Fig 1C). This suggests substantial interindividual variability in the sensitivity of hair matrix keratinocytes to switch on the apoptotic machinery, e.g. as a reflection of the local balance of Bcl-2 and Fas expression in the hair matrix (Müller-Röver et al, 1999; Sharov et al, 2004; Sharova

et al, 2014) following the mitosis-targeting damage inflicted by paclitaxel. This could correspond to the highly variable severity of hair loss seen in the clinic in response to identical chemotherapy regimens (Chung et al, 2013; Paus et al, 2013).

To dissect and model the early effects of taxanes on the human hair follicle beyond the initial 24-h treatment period, hair follicles treated with paclitaxel and docetaxel were washed out of drug-containing medium and permitted to continue in organ culture for an additional 24- to 48-h period. Analysis at this time point showed consistent and sustained increases in the number of pH3$^+$ and cleaved caspase-3$^+$ cells in the hair matrix (Appendix Fig S1A–F), indicating lasting hair follicle cytotoxicity imposed by taxanes even after drug washout.

**Taxanes induce the accumulation of cleaved caspase-3$^+$ and pH3$^+$ cells within the stem/progenitor-rich outer root sheath**

Hair follicle epithelial stem/progenitor cell damage has never been documented for taxane chemotherapy, yet would plausibly explain

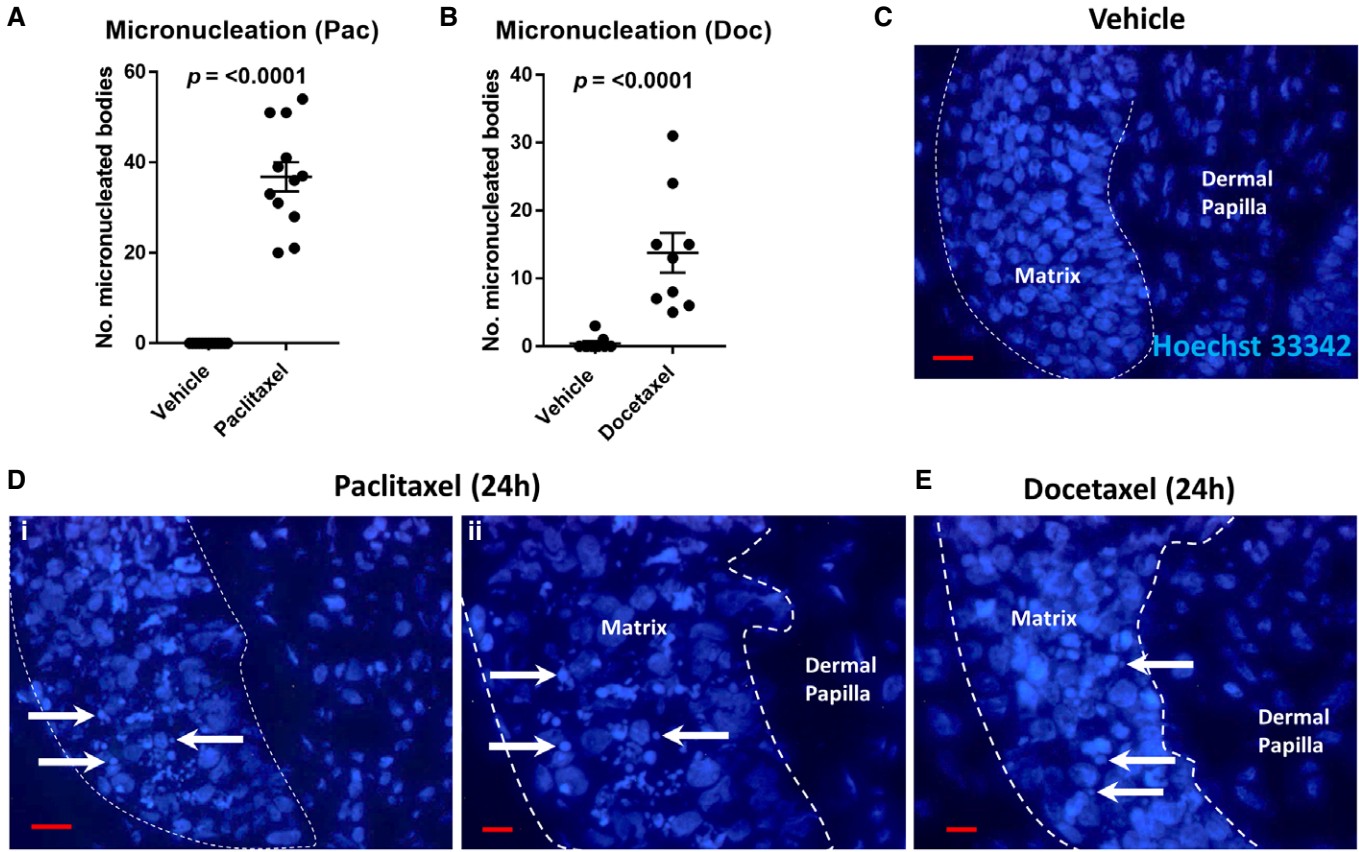

**Figure 2. Taxanes induce micronucleation in the human anagen hair follicle matrix.**

A, B   The presence of micronucleated cells in the hair matrix in paclitaxel- and docetaxel-treated (100 nM, 24 h) hair follicles (HF) compared to vehicle is significant ($P \leq 0.0001$). Mann–Whitney *U* test performed using *N* of 12–13 HFs (paclitaxel) and 8 HFs (docetaxel) from three patients. Error bars are standard error of the mean.

C   Hoechst 33342 staining of healthy cell nuclei comprising the hair matrix (lined) and dermal papilla in untreated (vehicle) human HFs. 20-μm scale.

D   Paclitaxel treatment (100 nM, 24 h) induces the formation of micronucleated bodies, as visualised by Hoechst 33342 staining (arrows), localising to the proliferative region of the hair matrix. i—20-μm scale; ii—10-μm scale.

E   100 nM docetaxel treatment was also seen to promote the formation of micronucleated bodies (arrows). 10-μm scale.

Source data are available online for this figure.

the permanency of hair loss in taxane chemotherapy-induced alopecia (Paus *et al*, 2013; Gao *et al*, 2019). Therefore, we next investigated the effect of anti-mitotic taxane chemotherapy on the proliferative, yet slower cycling, stem/progenitor cell-containing outer root sheath of human anagen VI scalp hair follicles (Purba *et al*, 2017b). We found that treatment of hair follicles with paclitaxel or docetaxel significantly increased the number of cleaved caspase-3[+] and pH3[+] cells in the outer root sheath (Appendix Fig S1G–K). Dual immunofluorescence staining for cleaved caspase-3 or pH3, alongside the hair follicle epithelial stem/progenitor cell marker keratin 15 (K15) (Cotsarelis, 2006; Purba *et al*, 2014) in paclitaxel-treated hair follicles, showed that accumulating cleaved caspase-3[+] and pH3[+] cells localise within and immediately adjacent to K15[+] expressing cells of the bulge stem cell region and proximal bulb outer root sheath progenitor compartment (Fig 4Ai–ii) (Purba *et al*, 2014, 2015).

An analysis of the number of cleaved caspase-3[+] cells following extended human hair follicle organ cultures (see Materials and Methods) revealed that paclitaxel significantly increases apoptosis in the K15[+] bulge (Fig 4B and C). Consistent with the effects

observed in the hair matrix, paclitaxel did not significantly affect the number of Ki-67[+] cells in the K15[+] bulge (Fig 4D) (proliferation in the bulge is enhanced during *ex vivo* organ culture; Purba *et al*, 2017b). In addition, γH2A.X analysis (Mah *et al*, 2010) also showed that paclitaxel treatment significantly increases DNA damage in the K15[+] bulge (Fig 4E and F), possibly because of prolonged mitotic arrest (Ganem & Pellman, 2012).

Together, these data provide the first evidence that proliferating stem/progenitor cell populations of human anagen VI hair follicles located in distinct compartments of the outer root sheath (Purba *et al*, 2017b) are indeed damaged by taxane chemotherapy, at least under *ex vivo* conditions. This damage could play a pivotal role in the pathobiology of permanent taxane-induced alopecia and calls for the rapid development of hair follicle stem cell-protective strategies in the management of this form of chemotherapy-induced alopecia (Paus *et al*, 2013) to curb the alarming rise in reported cases (Prevezas *et al*, 2009; Tallon *et al*, 2010; Miteva *et al*, 2011; Palamaras *et al*, 2011; Kluger *et al*, 2012; Tosti *et al*, 2013; Kang *et al*, 2018; Martín *et al*, 2018).

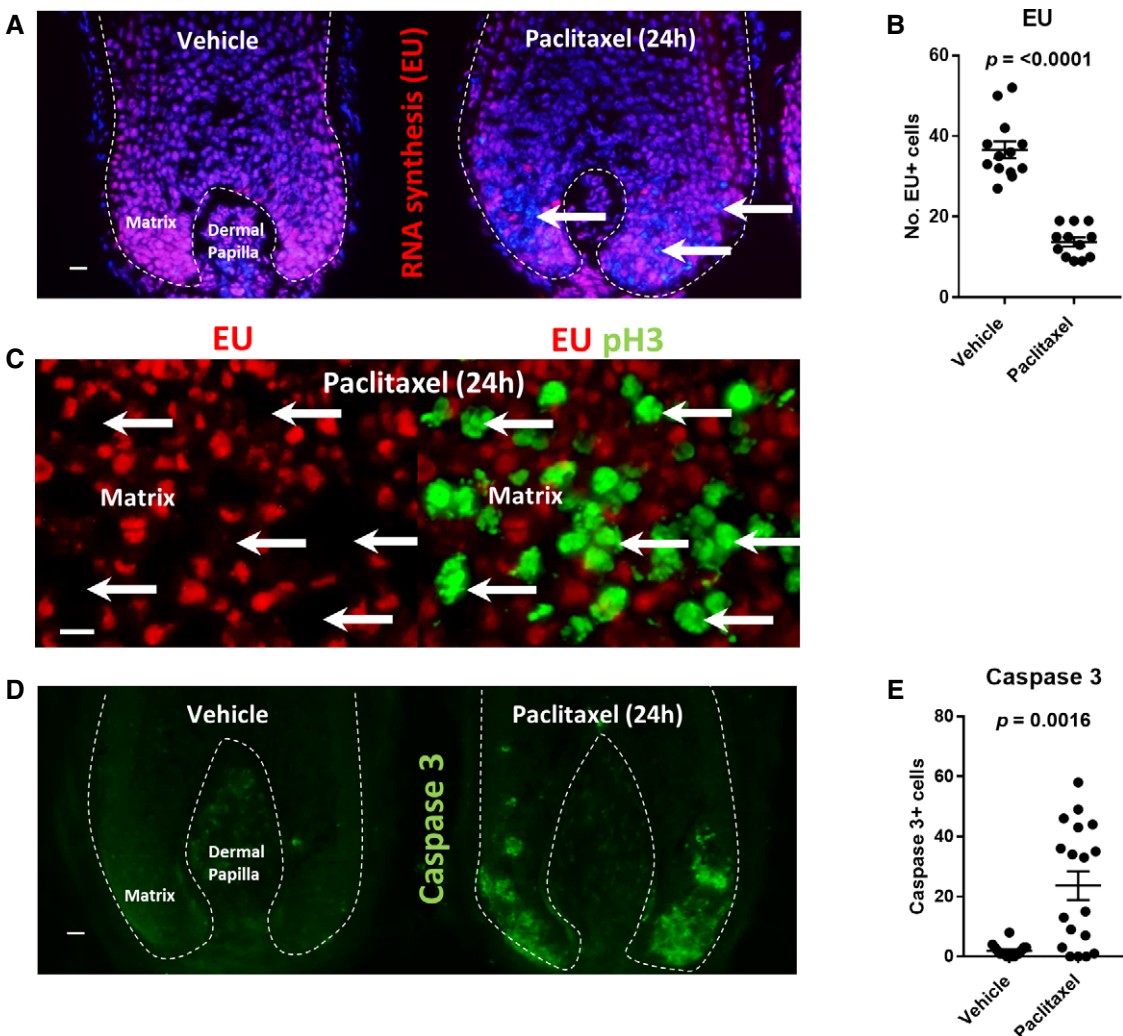

**Figure 3. Paclitaxel blocks nascent transcription and significantly increases cleaved caspase-3 immunoreactivity in hair matrix keratinocytes.**

A Nascent RNA synthesis, as detected by ethynyl uridine (EU) incorporation, is blocked within clusters of nuclei in the hair matrix (arrows) following paclitaxel treatment (100 nM, 24 h). 20-μm scale.

B Quantitative analysis highlights a significant ($P ≤ 0.0001$) decrease in the number of EU$^+$ nuclei following 24-h paclitaxel treatment. Welch's $t$-test performed using $N$ of 11–12 hair follicles (HFs) from three patients.

C Representative dual fluorescence stain highlights how EU incorporation in the hair matrix is blocked within the pH3$^+$ cell population that accumulates in response to paclitaxel treatment (see Fig 1). 10-μm scale.

D Cleaved caspase-3 expression in the hair matrix following 24-h paclitaxel treatment. 20-μm scale.

E 100 nM paclitaxel treatment significantly ($P = 0.0016$) increases the number of cleaved caspase-3$^+$ cells in the hair matrix after 24 h. Mann–Whitney $U$ test performed using $N$ of 16–18 HFs from five patients.

Data information: Values plotted represent the mean number of positive cells counted per HF analysed. Error bars are standard error of the mean.

Source data are available online for this figure.

## Targeted pharmacological inhibition of CDK4/6 induces G1 arrest in proliferating human hair matrix keratinocytes *ex vivo*

We next aimed to identify a suitable small molecule capable of potently and specifically inducing cell cycle arrest in matrinocytes during hair follicle organ culture that could be employed to counteract the mitosis-targeting cytotoxicity of taxanes documented above. In this context, cell cycle arrest therapy has previously been advocated, but a prominent paper proposing this strategy in a rodent model of chemotherapy-induced alopecia (Davis *et al*, 2001) was later withdrawn (Davis *et al*, 2002). Therefore, proof-of-principle for this potential chemotherapy-induced alopecia management strategy remains to be demonstrated, namely in human scalp hair follicles.

Arguing that arresting proliferating hair matrix keratinocytes and stem/progenitor cells in the G1 phase of the cell cycle should sharply reduce hair follicle keratinocyte vulnerability to mitosis-targeting taxane cytotoxicity, we turned to the small-molecule palbociclib, a highly specific inhibitor of the G1 progression kinases CDK4/6, which results in a reversible cell cycle phase-specific arrest in G1 (Fry *et al*, 2004).

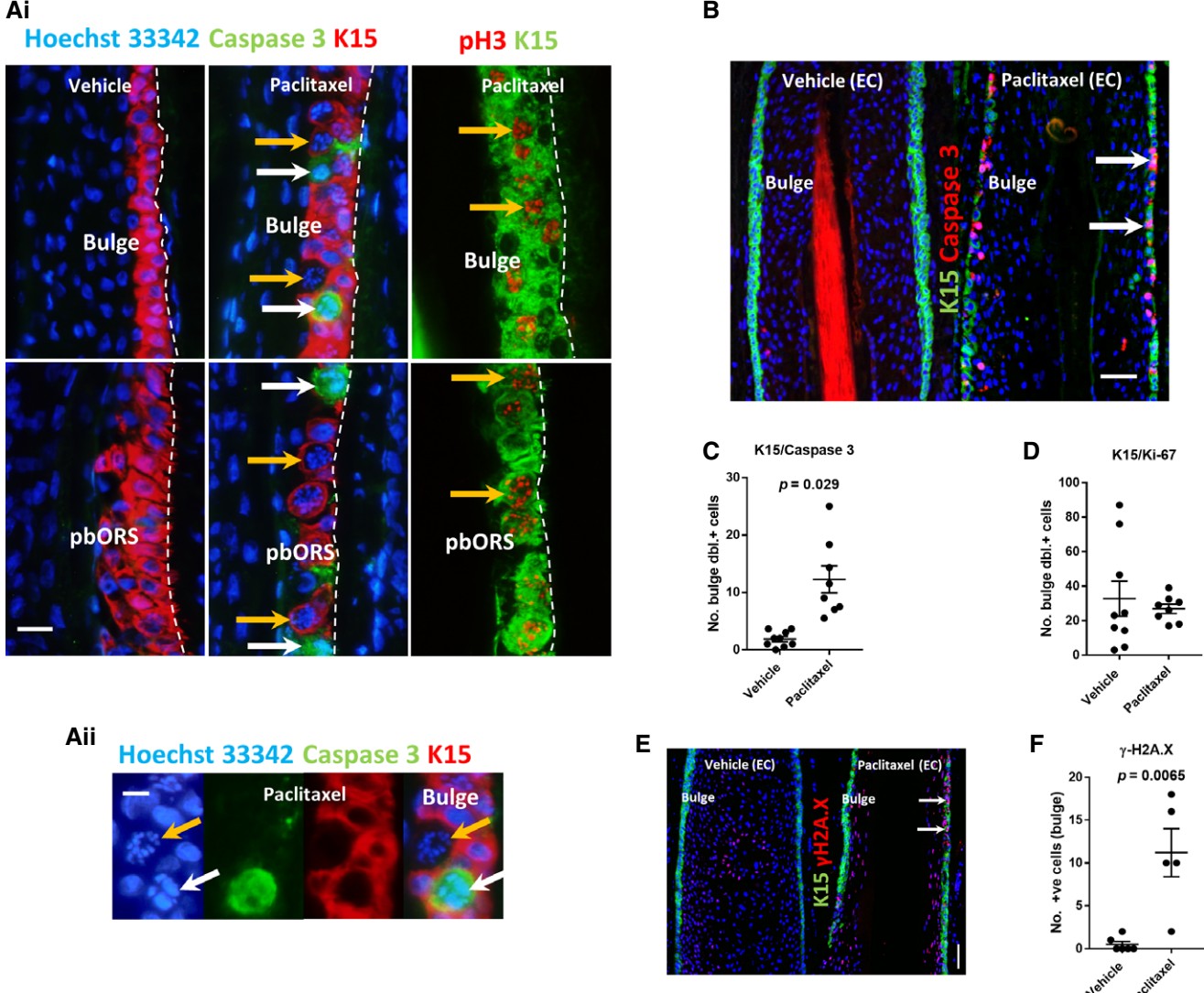

**Figure 4. Taxanes induce apoptosis and mitotic defects within human hair follicle K15+ epithelial stem/progenitor cell niches.**

A   Paclitaxel treatment (Ai) promotes mitotic arrest (orange arrows/pH3+ cells) and apoptosis (white arrows/caspase-3+ cells), within the K15+ bulge and K15+ proximal bulb outer root sheath (pbORS) stem/progenitor cell compartments of the human hair follicle (HF). 20-μm scale. (Aii) High magnification montage demonstrating mitotic arrest (orange arrow) and apoptosis/caspase-3 positivity (white arrow) within the K15+ bulge. 10-μm scale.

B   Representative immunofluorescence images of heightened cleaved caspase-3 immunoreactivity (arrows) within the K15+ bulge following extended paclitaxel organ culture experiments (see Materials and Methods). 50-μm scale.

C   Graph showing significantly (P = 0.029) increased K15/caspase-3 double-positive cells within the bulge following extended paclitaxel HF organ cultures. Welch's t-test performed using N of 8–9 HFs from three patients.

D   K15+ cells of the human HF bulge express Ki-67 during extended organ culture experiments. Paclitaxel treatment did not significantly affect the number of bulge K15/Ki-67 double-positive cells. Unpaired t-test performed using N of 8–9 HFs from three patients.

E   Representative double immunofluorescence images of elevated γH2A.X immunoreactivity (arrows) within the K15+ bulge following extended paclitaxel organ culture experiments (see Materials and Methods). 50-μm scale.

F   γH2A.X analysis showing a significant (P = 0.0065) increase in the number of cells with DNA double-strand breaks in the K15+ bulge following extended organ culture experiments. Mann–Whitney U test performed using N of 5–6 HFs from two patients.

Data information: Values plotted represent the mean number of positive cells counted per HF analysed. Error bars are standard error of the mean. pbORS, proximal bulb outer root sheath. EC, extended cultures.
Source data are available online for this figure.

Treatment of human hair follicles *ex vivo* with 1 μM palbociclib for 24 h resulted in a blockade of DNA synthesis, as marked by a significant decrease in the number of matrix keratinocytes that had

incorporated EdU *in situ* (Appendix Fig S2A). This showed the successful induction of cell cycle arrest through the stalling of G1 progression. In accordance with the induction of a G1-specific

arrest, the total number of cycling cells (Ki-67[+]) was reduced within the hair matrix (Appendix Fig S2B). In addition, the fraction of mitotic cells, as marked by pH3 staining, was significantly reduced by CDK4/6 inhibition (Appendix Fig S2C). Importantly, 24-h palbociclib treatment alone did not increase the number of cleaved caspase-3[+] cells (Appendix Fig S2D). Together, these data show that short-term administration of palbociclib is not cytotoxic to human scalp hair follicles and successfully arrests G1/S progression in hair matrix keratinocytes (Appendix Fig S2E).

### CDK4/6 inhibition blocks the cytotoxic effects of paclitaxel in the human hair follicle matrix

Next, we probed how pharmacologically imposed G1 cell cycle arrest influences human hair follicle responses to paclitaxel treatment. Hair follicles were first pre-incubated with 1 μM palbociclib for a period of 18 h. This incubation period was chosen to account for the calculated time it would take for post-G1 human hair matrix keratinocytes to complete a single round of proliferation through S-G2-M and exit the cell cycle (or re-enter G1) (Weinstein & Mooney, 1980). Following this pre-incubation step, hair follicles were then exposed to both paclitaxel and (reapplied) palbociclib for a further 24 h, after which hair follicles were immediately isolated at the 42 h time point (alongside vehicle; palbociclib-only; and paclitaxel-only treatment groups; Fig 5A).

Palbociclib-only treatment for a period nearing 2 days caused a dramatic reduction in Ki-67 expression within matrix keratinocytes (Fig 5B), well beyond the decreases seen after 24 h of treatment (Appendix Fig S2B). This result is consistent with the report that proteasome-mediated degradation of Ki-67 occurs following CDK4/6 inhibition and G1 arrest (Sobecki et al, 2017). The elimination of Ki-67 expression in matrix keratinocytes by CDK4/6 inhibition was also apparent in the dual palbociclib/paclitaxel treatment group (Fig 5B).

Consequently, pharmacological G1 arrest blocked the abnormal accumulation of pH3[+] cells in the hair matrix that is otherwise strongly induced in paclitaxel-only treated hair follicles (Fig 5C). In accordance with this, palbociclib pre-treatment also prevented an increase in the number of cleaved caspase-3[+] cells within the hair follicles of a donor sensitive to early paclitaxel-induced apoptosis (Fig 5D).

Together, these in situ data, dissected within a complex and highly proliferative human mini-organ, show that the mitosis-targeting effects of paclitaxel chemotherapy are rendered ineffective by targeted G1 phase-specific arrest using the CDK4/6 inhibitor palbociclib (Fig 5Ei–iii). These results therefore provide a proof-of-principle that therapeutic cell cycle arrest approaches have the potential to protect hair matrix keratinocytes from taxanes and thus prevent hair follicle damage leading to chemotherapy-induced alopecia.

### Palbociclib-induced G1 arrest is reversible in the hair matrix

We next probed the reversibility of CDK4/6 inhibition. To achieve this, we repeated hair follicle organ cultures as defined above with the addition of a drug washout step at 42 h and a subsequent 24- to 48-h culture period free from palbociclib (Fig 6A).

In the hair matrix of palbociclib-only treated hair follicles, EdU incorporation resumed in the majority (66%) of cases following drug washout (Fig 6B; Appendix Fig S3A). This demonstrates that CDK4/6 inhibition is reversible (Fry et al, 2004) in human hair follicle matrix keratinocytes. Furthermore, palbociclib-only treated hair follicles did not show increased cleaved caspase-3 immunoreactivity (Fig 6C; Appendix Fig S3B), indicating that transient G1 arrest is not cytotoxic to hair matrix keratinocytes.

### Paclitaxel requires active proliferation to exert hair matrix cytotoxicity and may be retained in the hair follicle

We then analysed hair follicles treated with both palbociclib and paclitaxel, after washout and culture in drug-free medium (Fig 6A), and found that DNA synthesis also resumed within the hair matrix in this treatment group (Fig 6B). This resumption of cell cycle progression was met with a paclitaxel-induced accumulation of pH3[+] cells within the hair matrix (Fig 6D). This shows that paclitaxel can exert an anti-mitotic effect as soon as the cell cycle resumes and progression to mitosis is permitted (Fig 6E). This effect occurred despite the removal of paclitaxel from the culture medium, which suggests drug retention of paclitaxel in hair follicle keratinocytes, or otherwise could be indicative of existing damage that only becomes apparent following resumed cell proliferation. Notably, paclitaxel has been described to be retained in cancer cells for 1 week in vitro and in vivo (Mori et al, 2006).

Resumed cell proliferation in the combined palbociclib and paclitaxel treatment group, post-drug washout, resulted in a trending (yet not significant) increase in the number of cleaved caspase-3[+] cells in the hair matrix (Fig 6C). The lack of statistical significance in this treatment group post-washout, in contrast to paclitaxel-only treated hair follicles, can be partly attributed to the lack of a G1 arrest reversal in the hair matrix of a small number of hair follicles. Indeed, in the few hair follicles treated with palbociclib and paclitaxel that showed little to no EdU incorporation following drug washout, no corresponding accumulation of cleaved caspase-3[+] or pH3[+] cells was observed (Appendix Fig S4). Together, these results show how cell proliferation is necessary for paclitaxel to exert any anti-mitotic and pro-apoptotic effects within the hair matrix and that paclitaxel may be retained in hair follicle keratinocytes.

### Paclitaxel and palbociclib do not promote catagen in human hair follicles during ex vivo organ culture

Hair follicles respond to chemotherapy with distinct changes in hair follicle cycling, i.e. either by prolonged maintenance in anagen ("dystrophic anagen" pathway) or by rapid, premature induction of apoptosis-driven hair follicle regression ("dystrophic catagen" pathway; Paus et al, 1994, 2013; Hendrix et al, 2005). To determine therefore how palbociclib and paclitaxel treatments influence the hair cycle during ex vivo culture, we staged treated hair follicles at the indicated 42 h and 66–90 h time points (Appendix Fig S5). Morphological analysis showed that paclitaxel treatment did not promote hair follicles to enter catagen (Appendix Fig S5). This indicates that, despite massive paclitaxel-induced mitotic defects and apoptosis, taxane chemotherapy initially promotes the "dystrophic anagen" pathway (Paus et al, 2013).

Palbociclib-only treatment also did not result in catagen induction ex vivo, perfectly in line with the observation that palbociclib does not induce apoptosis in the hair matrix (Fig 6C; Appendix Fig S2D). This demonstrates that palbociclib can be effectively used to

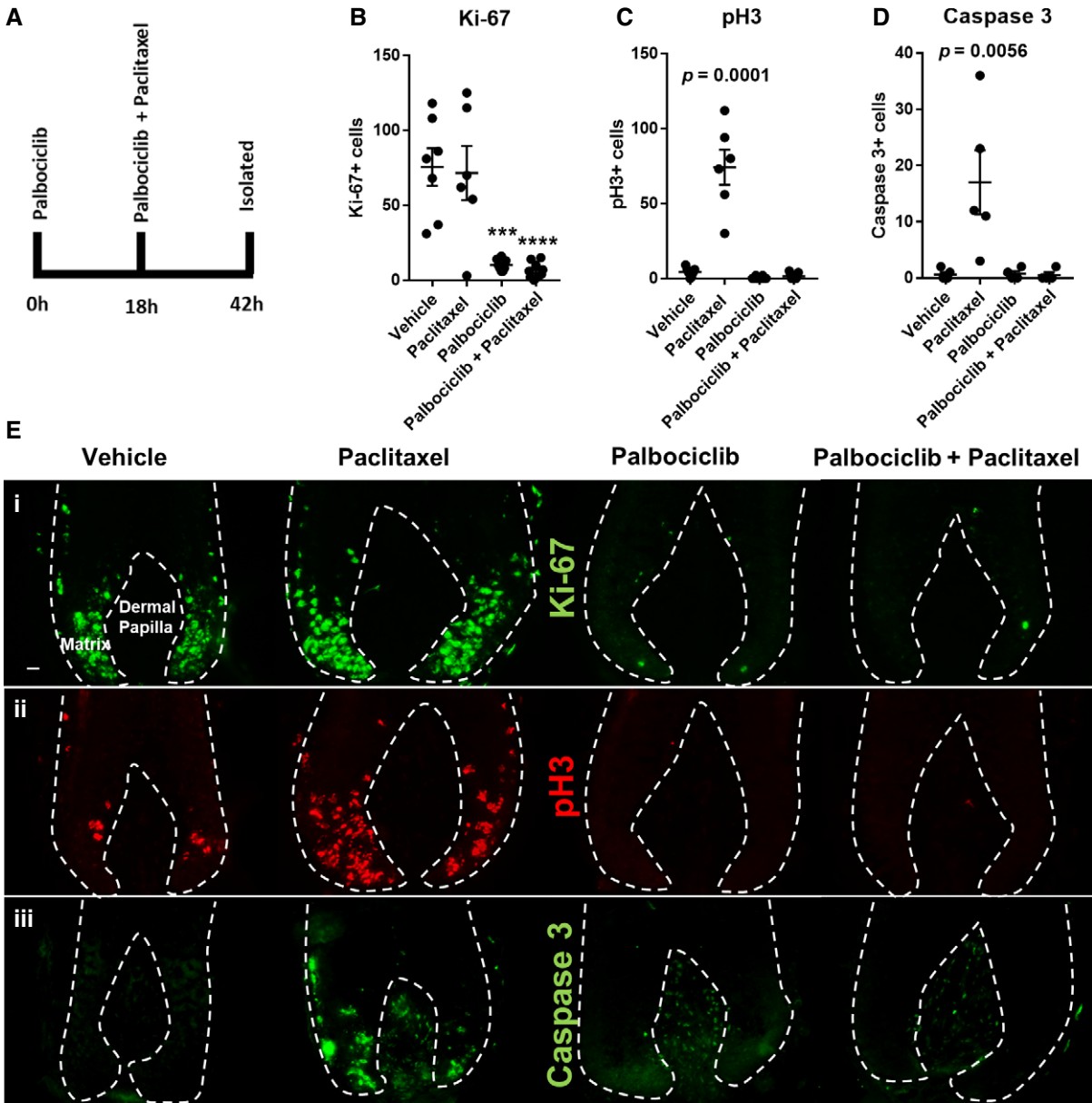

**Figure 5. Palbociclib blocks paclitaxel-induced mitotic defects and apoptosis in the human hair follicle matrix.**

A Schematic of experimental design. Hair follicles (HFs) were pre-incubated with the CDK4/6 inhibitor palbociclib for 18 h, followed by a further incubation period with and without paclitaxel (or paclitaxel alone) for an additional 24 h.

B Palbociclib-only- and palbociclib plus paclitaxel dual-treated HFs show marked reductions in Ki-67 expression in the hair matrix beyond that seen with just 24-h treatment (see Appendix Fig S2B). Data also confirm that Ki-67 expression is unaffected by paclitaxel treatment (see also Fig 1A). Analysis performed using N of 6–9 HFs per condition from three patients. Ordinary one-way ANOVA with multiple comparisons performed. Adjusted P values = 0.0001[***] and 0.0001 [****], respectively.

C Data confirming that paclitaxel treatment significantly increases (adjusted P value = 0.0001) the number of pH3[+] cells in the hair matrix (see also Fig 1C). This effect was not observed when paclitaxel-treated hair follicles were pre- and co-incubated with palbociclib. Analysis performed using N of 6–9 HFs per condition from three patients. Ordinary one-way ANOVA with multiple comparisons performed.

D Cleaved caspase-3 analysis in hair follicles from a patient donor that showed apoptotic sensitivity to paclitaxel treatment (see Fig 3D and E and results, main text) (adjusted P value = 0.0056). Paclitaxel-treated hair follicles pre- and co-incubated with palbociclib do not show enhanced cleaved caspase-3 expression, contrasting with paclitaxel-only treatment. Ordinary one-way ANOVA with multiple comparisons performed using N of 4–5 HFs per condition.

E Immunofluorescence data representing how palbociclib and/or paclitaxel affect (i) Ki-67 expression, (ii) pH3 immunoreactivity and (iii) cleaved caspase-3 expression. 20-μm scale.

Data information: Values plotted represent the mean number of positive cells counted per HF analysed. Error bars are standard error of the mean.
Source data are available online for this figure.

 

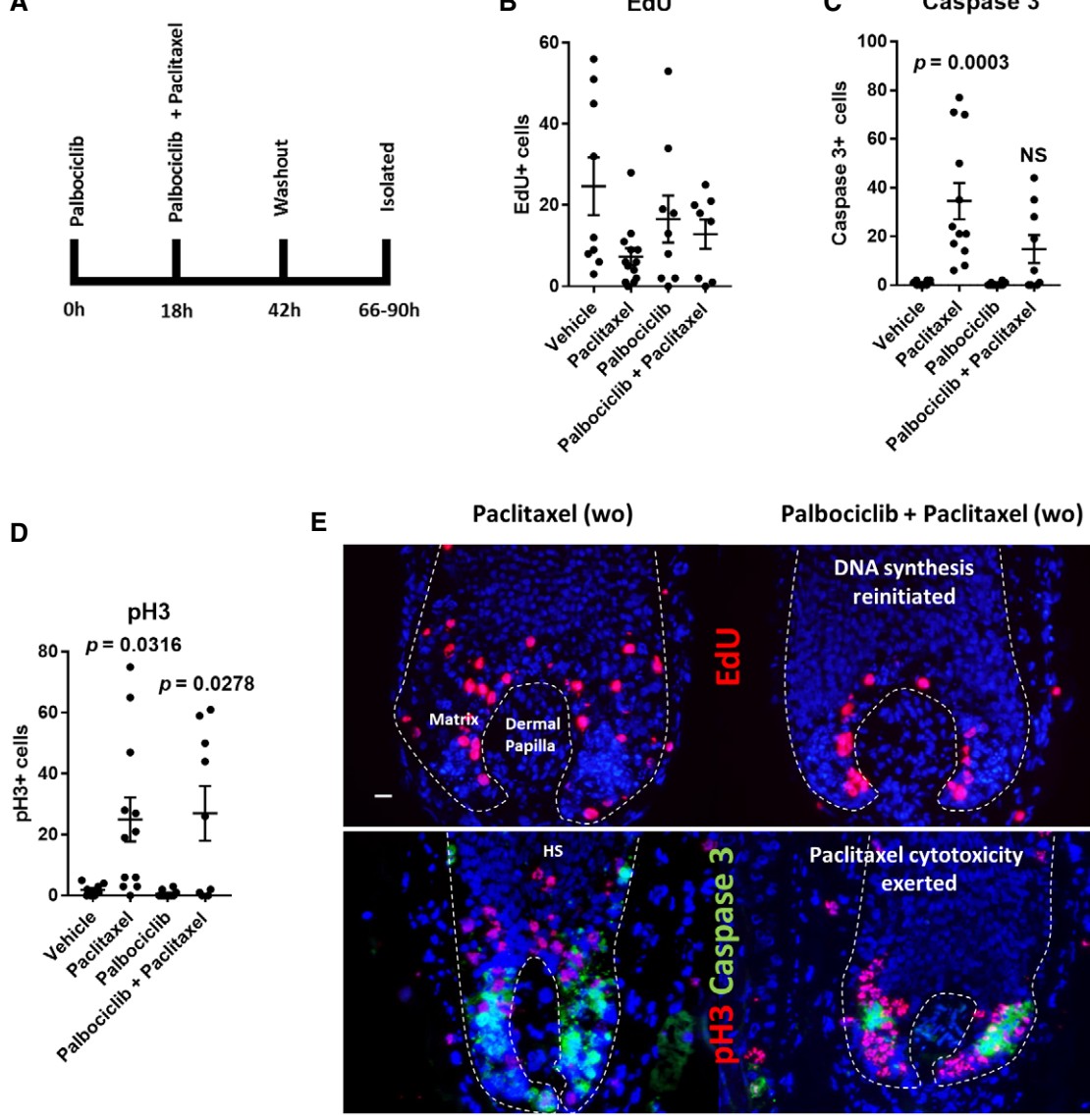

**Figure 6. Transient CDK4/6 inhibition emphasises the cell cycle dependency of paclitaxel cytotoxicity in the human hair follicle.**

A   Schematic of experimental design. Hair follicles (HFs) were pre-treated with palbociclib for 18 h and then further incubated either with or without paclitaxel for a further 24 h. Subsequently, drug-containing media was washed out and HFs were left in organ culture for an additional 24- to 48-h period.

B   No significant differences in EdU incorporation were found in any experimental conditions compared to vehicle. DNA synthesis in the hair matrix resumed in the palbociclib-only- and palbociclib plus paclitaxel dual-treated conditions following washout (see also Appendix Fig S3). Ordinary one-way ANOVA with multiple comparisons performed using N of 8–13 HFs from three patients.

C   A trending increase in the number of cleaved caspase-3⁺ cells was seen in the hair matrix of HFs dual treated with paclitaxel and palbociclib following drug washout, but this was not significant, whereas paclitaxel-only treatment significantly increased the number of cleaved caspase-3⁺ cells in the hair matrix (adjusted *P* value = 0.0003). Palbociclib-only treatment did not increase the number of cleaved caspase-3⁺ cells in the hair matrix following drug washout. Ordinary one-way ANOVA with multiple comparisons performed using N of 8–12 HFs from three patients. Caspase-3 vehicle vs. paclitaxel data are also used in Appendix Fig S1E.

D   HFs treated with paclitaxel alone showed a significant (adjusted *P* value = 0.0316) increase in the number of pH3⁺ cells in the matrix after washout. A significant (adjusted *P* value = 0.0278) increase in the number pH3⁺ cells was seen in HFs dual treated with both paclitaxel and palbociclib following drug washout. Palbociclib treatment alone showed no significant change in the number of pH3⁺ hair matrix keratinocytes after drug washout. Ordinary one-way ANOVA with multiple comparisons performed using N of 8–12 HFs from three patients. pH3 vehicle vs. paclitaxel data are also used in Appendix Fig S1B.

E   Representative fluorescence images of EdU and pH3/cleaved caspase-3 staining in paclitaxel-only- and palbociclib plus paclitaxel dual-treated HFs following drug washout. Where DNA synthesis has resumed following reversal of the G1 arrest, paclitaxel cytotoxicity is emergent. 20-μm scale.

Data information: Values plotted represent the mean total number of cells counted per HF analysed. Error bars are standard error of the mean. Please also see supporting data in Appendix Fig S3 and S4. HS, hair shaft; wo, washout.

Source data are available online for this figure.

induce a reversible arrest that is non-toxic and does not promote catagen, and thus permits growth to resume in anagen upon discontinuation of drug application (as suggested by resumed EdU incorporation in the hair matrix after palbociclib removal; Fig 6B). Therefore, temporary G1 arrest therapy by palbociclib is unlikely to promote (clinically undesired) catagen development and the associated subsequent inevitable hair loss (telogen effluvium).

**Pharmacological G1 arrest protects proliferating epithelial stem/progenitor cell compartments from taxane-induced cytotoxicity**

This left to be explored how CDK4/6 inhibition influences the response of proliferative epithelial stem and progenitor cells in the stem/progenitor-rich outer root sheath (Cotsarelis, 2006; Garza et al, 2011; Purba et al, 2017b) to taxane chemotherapy. To do this, we repeated the experimental conditions as defined in Figs 5A and 6A, this time focusing our analysis on compartments of the outer root sheath.

We found that, after isolation at the 42 h time point (without washout) (Appendix Fig S6A), proliferating outer root sheath keratinocytes were sensitive to pharmacologically induced cell cycle arrest, signified by a profound loss of Ki-67[+] outer root sheath keratinocytes following palbociclib treatment (Appendix Fig S6B). In turn, palbociclib pre- and co-treatment blocked the paclitaxel-induced accumulation of pH3[+] cells in the outer root sheath after 24 h (Appendix Fig S6C). These results are consistent with findings within the hair matrix under the same experimental conditions (Fig 5).

We then investigated how palbociclib influenced paclitaxel-induced outer root sheath damage after washout and culture in drug-free medium (Fig 7A). In contrast to the matrix, no hair follicles in the palbociclib-only treatment group showed any EdU incorporation into the outer root sheath after drug washout (Fig 7B). On the other hand, 50% of hair follicles dual treated with palbociclib and paclitaxel displayed some, albeit minimal, EdU incorporation in the outer root sheath (Fig 7B), which could represent outer root sheath cell proliferation stimulated by chemotherapy-induced damage (Huang et al, 2017).

Hair follicles treated with both palbociclib and paclitaxel showed no significant increases in cleaved caspase-3 and pH3 in the outer root sheath following drug washout, which contrasted markedly with paclitaxel-only treatment (Fig 7C and D). These results are a consequence of very limited proliferation in the outer root sheath following palbociclib treatment, which blocks the cytotoxicity of paclitaxel (Fig 7E and F). Within the few hair follicles that did show DNA synthesis in the outer root sheath following drug washout, paclitaxel promoted the accumulation of pH3[+] cells (Appendix Fig S7). Double-staining of K15 with either cleaved caspase-3 or pH3 revealed that outer root sheath cell cycle arrest through palbociclib treatment antagonises paclitaxel-induced mitotic defects and apoptosis within K15[+] stem/progenitor niches (Fig 7G and H).

Together, these data emphasise the cell cycle-dependent effects of paclitaxel treatment on the outer root sheath, mirroring the effects of paclitaxel on hair matrix keratinocytes. Furthermore, these data show a proof-of-principle for the therapeutic application of cell cycle arresting therapies to protect proliferative stem/progenitor-rich compartments of the hair follicle outer root sheath, whose destruction and exhaustion via chemotherapy is likely to promote the pathogenesis of severe and potentially permanent chemotherapy-induced alopecia (Paus et al, 2013).

# Discussion

Here, we elucidate how taxanes damage the human scalp hair follicle epithelium, both in the rapidly proliferating anagen hair matrix and in stem/progenitor cell-rich outer root sheath niches, using a new, clinically relevant full-length hair follicle organ culture model optimised for the ex vivo study of paclitaxel and docetaxel. Namely, we show how taxanes induce massive mitotic defects and apoptosis in anagen hair matrix keratinocytes signified by an accumulation of pH3[+] (i.e. mitotically arrested) cells, micronucleation, transcriptional arrest and cleaved caspase-3[+] cells. Moreover, we provide the first evidence that taxanes also damage the stem/progenitor-rich outer root sheath, including within defined K15[+] hair follicle compartments (Purba et al, 2014; 2015).

We also provide the first proof-of-principle that arresting hair matrix keratinocytes and hair follicle epithelial stem/progenitor cells in G1, using the CDK4/6-inhibitor palbociclib, provides relative protection from taxane-induced hair follicle damage without promoting catagen development or exerting additional hair follicle toxicity. Together, this not only sheds much-needed light on the previously unknown pathobiology of taxane chemotherapy-induced alopecia and explains why this hair loss can be permanent, but also identifies pharmacological G1 arrest as a plausible and promising strategy for managing taxane chemotherapy-induced alopecia. It is noteworthy that scalp cooling, the only approved treatment currently available to protect against chemotherapy-induced alopecia (Friedrichs & Carstensen, 2014; Cigler et al, 2015; Rugo et al, 2017; Rice et al, 2018), may, in part, work through the deceleration of cell cycle progression in proliferating hair follicle keratinocytes (Rieder & Cole, 2002; Al-Tameemi et al, 2014). Thus, it is tempting to probe next whether the efficacy of scalp cooling can be further enhanced by combining it with pharmacological G1 arrest.

CDK4/6 has been reported to be a promising target to protect tissues and organs from the cytotoxicity of chemotherapy (i.e. liver, kidney; DiRocco et al, 2014; Pabla et al, 2015; He et al, 2017). However, the suitability of targeting CDK4/6 to protect against chemotherapy damage in the hair follicle and prevent chemotherapy-induced alopecia in patients (i.e. via topical administration) remains to be fully elucidated in vivo. For any such pharmacological cell cycle arrest approach to be successful, the candidate therapeutic would need to satisfy several criteria. Namely, the ideal candidate agent would need to be: (i) topically applicable (so as to circumvent the protection of tumour cells from chemotherapy by cell cycle arrest, whilst greatly reducing the recognised adverse effects of systemically administered cell cycle inhibitors like palbociclib; Ro et al, 2015; Finn et al, 2016), (ii) fast acting; (iii) reversible; (iv) non-cytotoxic; (v) should not promote catagen (which would lead to telogen effluvium (Paus & Cotsarelis, 1999; Paus et al, 2013)), and (vi) should permit the resumption of hair growth in anagen.

Whilst transient CDK4/6 inhibition through palbociclib treatment is not cytotoxic to the hair follicle and does not promote catagen, we found that matrix proliferation, vital for hair growth, did not resume in a minority of hair follicles. This effect may not be apparent in vivo and does not preclude a CDK4/6 targeting approach from being further optimised (e.g. dosage) for clinical application. But this does highlight the challenges one might face when effectively

"fighting fire with fire", e.g. risking the inadvertent promotion of a hair growth arrest, or even the promotion of cell senescence (Klein *et al*, 2018) in the hair follicle.

Systemic palbociclib therapy to treat hormone receptor-positive HER2-negative breast cancer (once daily for weeks) has unsurprisingly been found to cause hair loss (Ro *et al*, 2015; Finn *et al*, 2016). This underscores that transient, short-term intrafollicular G1 arrest by topical application of CDK4/6 inhibitors, rather than prolonged G1 arrest by systemic agents, is the most promising strategy to obtain the desired anti-chemotherapy-induced alopecia effect.

Furthermore, another key challenge that future, cell cycle arrest-based protective strategies against taxane chemotherapy-induced alopecia must overcome relates to the length of time that paclitaxel is retained in tissues (Mori *et al*, 2006), which may limit the effectiveness of protective strategies. Indeed, given the delayed cytotoxic paclitaxel effects we observed after the reversal of cell cycle arrest following drug washout (Fig 6), our *ex vivo* data suggest that paclitaxel retention does occur in the human hair follicle (though pharmacodynamics in fully perfused hair follicles may obviously differ *in vivo*).

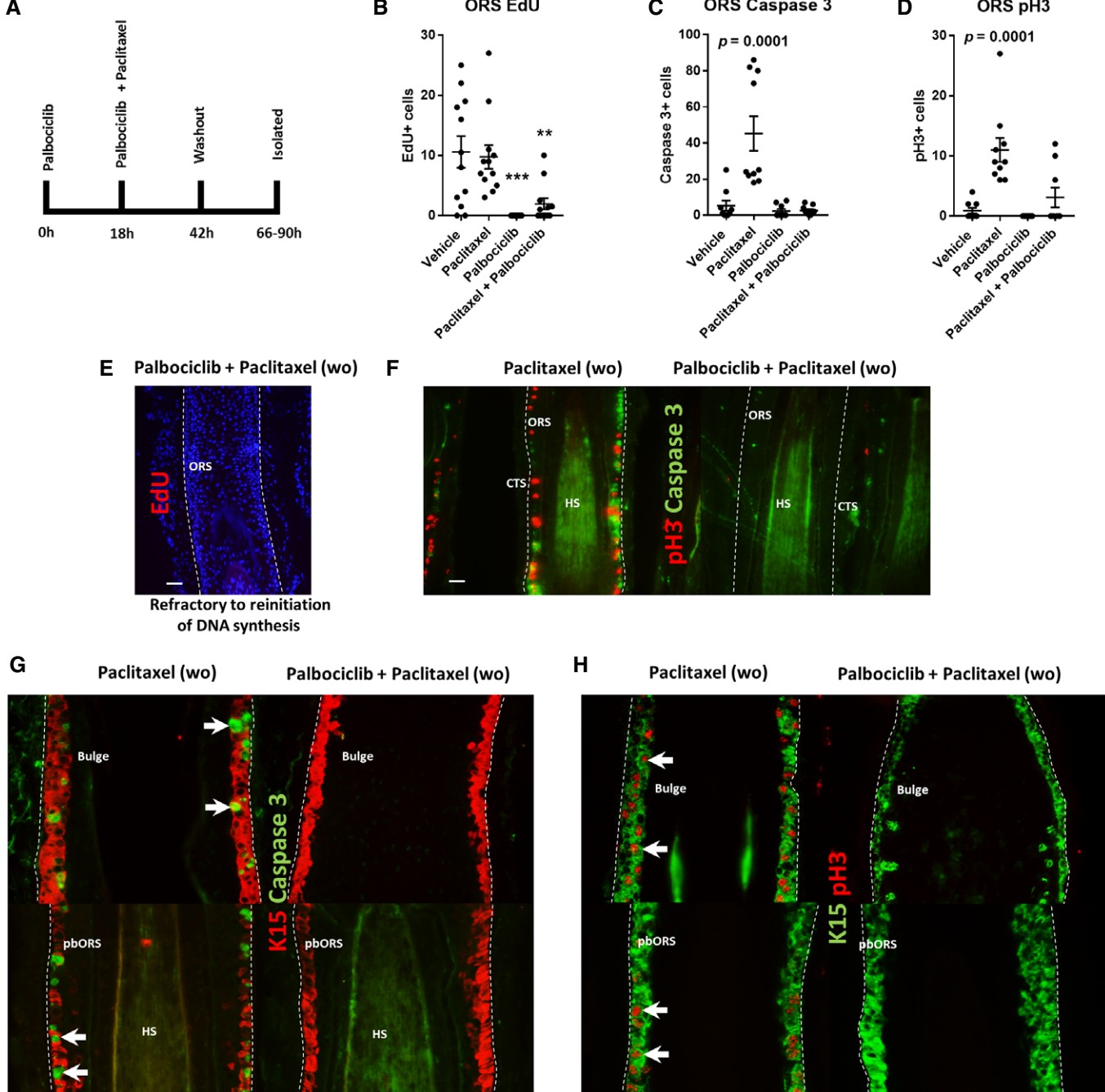

**Figure 7.**

◀

**Figure 7.  CDK4/6 inhibition antagonises paclitaxel cytotoxicity in the stem/progenitor cell-rich human hair follicle outer root sheath.**

A   Schematic of experimental design. Hair follicles (HFs) were pre-treated with (or without) palbociclib for 18 h and then incubated either with/without paclitaxel for a further 24 h. Subsequently, drug-containing media was washed out and HFs were cultured for an additional 24- to 48-h period.

B   Paclitaxel treatment did not affect the number of EdU$^+$ cells within the HF outer root sheath (ORS) following washout. In contrast, palbociclib-treated HFs lacked EdU incorporation following drug washout (adjusted P value = 0.0001 [***]). The number of EdU$^+$ cells was significantly decreased in the ORS of HFs dual treated with both paclitaxel and palbociclib (adjusted P value = 0.0021 [**]). Ordinary one-way ANOVA with multiple comparisons performed using N of 12–13 HFs from three patients.

C   HFs treated with paclitaxel showed a significant (adjusted P value = 0.0001) increase in the number of cleaved caspase-3$^+$ cells in the ORS following drug washout. Palbociclib-only and palbociclib plus paclitaxel dual-treated HFs showed no change in the number of cleaved caspase-3$^+$ ORS cells. Ordinary one-way ANOVA with multiple comparisons performed using N of 9–10 HFs from three patients. Data for vehicle vs. paclitaxel are utilised also in Appendix Fig S1H.

D   Paclitaxel-only-treated HFs showed a significant (adjusted P value = 0.0001) increase in the number of pH3$^+$ ORS cells following drug washout. Palbociclib treatment alone had no significant effect on the number of pH3$^+$ ORS cells. There was a trending, yet non-significant, increase in the number of pH3$^+$ ORS cells in HFs dual treated with palbociclib plus paclitaxel. Ordinary one-way ANOVA with multiple comparisons performed using N of 9–10 HFs from three patients. Data for vehicle vs. paclitaxel are utilised also in Appendix Fig S1I.

E   Representative image of ORS EdU incorporation staining in palbociclib plus paclitaxel dual-treated HFs following drug washout, whereby DNA synthesis has not resumed.

F   Representative images of pH3/caspase-3 staining within the ORS of paclitaxel-only- and palbociclib plus paclitaxel-treated HFs following drug washout. Please see supporting data in Appendix Fig S7.

G, H   Immunofluorescence images illustrating how palbociclib blocks the paclitaxel-induced accumulation of cleaved caspase-3$^+$ (G) and pH3$^+$ (H) cells in the K15$^+$ bulge and proximal bulb ORS (pbORS) compartments. K15/pH3 pbORS paclitaxel-only treatment HF image (H, bottom left panel) can be seen at a higher magnification in Fig 4Ai.

Data information: Values plotted represent the mean total number of cells counted per HF analysed. Error bars are standard error of the mean. CTS, connective tissue sheath; HS, hair shaft; pbORS, proximal bulb outer root sheath; wo, washout. (E, F) 50-μm scale; (G, H) 20-μm scale.

Despite these challenges and caveats, our data suggest that therapeutic cell cycle arrest approaches have the potential to prevent permanent taxane chemotherapy-induced alopecia. Permanent chemotherapy-induced alopecia following taxane treatment (Prevezas *et al*, 2009; Tallon *et al*, 2010; Miteva *et al*, 2011; Palamaras *et al*, 2011; Kluger *et al*, 2012; Tosti *et al*, 2013; Sibaud *et al*, 2016; Kang *et al*, 2018; Martín *et al*, 2018) is thought to arise in at-risk individuals from irreversible stem/progenitor cell destruction and proliferative exhaustion (Paus *et al*, 2013). In this study, we show that proliferating (yet relatively slower cycling) anagen hair follicle outer root sheath keratinocytes situated within epithelial stem/progenitor cell niches (Purba *et al*, 2017b) are indeed susceptible to taxanes. In turn, we show that G1 arrest via CDK4/6 inhibition attenuates paclitaxel cytotoxicity in proliferative epithelial stem/progenitor compartments. Therefore, these findings support that a topically delivered cell cycle arresting agent holds the potential to spare epithelial stem/progenitor cell sub-populations from taxane-mediated damage, thus preventing what is likely to be a key pathobiological event on the path to permanent chemotherapy-induced alopecia.

It is noteworthy that, over the course of fractionated, repetitive chemotherapy cycles, epithelial stem cells may be particularly vulnerable to exhaustion if they proliferate in response to damage (i.e. during dystrophic anagen or in the new anagen following dystrophic catagen; Paus *et al*, 2013). As our *ex vivo* test system is limited in addressing stem cell responses in this regard, models with xenotransplanted human scalp hair follicles (Oh *et al*, 2016) that permit long-term observations of chemotherapy effects *in vivo* (Yoon *et al*, 2016) are required to systematically follow-up the leads from the current work.

## Materials and Methods

### Hair follicle organ culture experiments

Hair follicles were microdissected from human occipital scalp tissue obtained from the Crown Clinic, Manchester, donated by patients under informed consent. Hair follicles were cultured within 24-well plates in 500 μl of hair follicle culture medium comprised of Williams E medium, 100 U/ml penicillin, 100 μg/ml streptomycin, 2 mM L-glutamine, 10 μg/ml insulin and 10 ng/ml hydrocortisone (Philpott *et al*, 1994; Langan *et al*, 2015). Upon isolation, hair follicles were randomly allocated to a given treatment group and cultured *ex vivo* for the periods indicated in the presence of compound dissolved in DMSO or DMSO-only (vehicle) at an equivalent concentration (max 0.01%). Compounds employed: 1 μM palbociclib (PD-0332991) (Selleckchem, #S1116), 100 nM paclitaxel (Taxol) (Tocris, #1097) and 100 nM docetaxel (Taxotere) (Abcam, #ab141248). Following culture, hair follicles were embedded in cryoembedding matrix and frozen using liquid nitrogen and stored at minus 80°C.

### Immunofluorescence

Pre-prepared 10 μm human hair follicle tissue cryosections on SuperFrost Plus slides (Thermo Scientific, #10149870) were fixed in ice-cold acetone for 10 min and allowed to dry at room temperature. ImmEdge Hydrophobic Barrier Pen (Vector labs #H-4000) was used to draw a barrier around individual tissue sections and allowed to dry. Tissue sections were treated overnight with primary antibody diluted in phosphate-buffered saline (PBS) within a humidified chamber kept at 4°C. Subsequently, excess solution was gently tapped off and tissue sections were washed twice with PBS. Sections were treated with fluorescent secondary antibody diluted 1:200 in PBS for 45 min at room temperature and protected from light. Slides were washed twice with PBS and stained with DAPI (1 μg/ml) for 2 min or, where indicated, Hoechst 33342 diluted in PBS (1:1,000) (10 mg/ml stock, Thermo Fisher) for 10 min at room temperature, protected from light. Slides were washed again in PBS and cover-slipped with aqueous mounting medium.

### Antibodies

Anti-Ki-67 [SP6] (Abcam, ab16667) (1:50), Phospho-Histone H3 (S10) (6G3) (Cell signalling, #9706) (1:100), Phospho-Histone H3

consistent exposure and zoom of a defined morphological reference area of interest (e.g. hair matrix). Fluorescence microscopy images were analysed using the cell counter plugin within ImageJ software (National Institutes of Health).

## Quantitative (immune-) histomorphometric and statistical analysis

The number of cells (double-) positive for a given read-out parameter was quantified within a consistently defined reference area (e.g. within the hair matrix or bulge) demarcated by morphology and/or the localisation of specific stem cell/proliferation markers on comparable hair follicle tissue sections. Values plotted on graphs represent the mean number of positive cells counted per hair follicle analysed. No power calculation was performed. Data were handled in GraphPad Prism 7 (GraphPad Software) for statistical analysis and graphical representation. Statistical testing employed the unpaired $t$-test for normally distributed data (or Welch's $t$-test for data with unequal variances), or the Mann–Whitney $U$ test. Normality testing was performed using D'Agostino–Pearson omnibus test within GraphPad. Analysis of data with multiple groups i.e. > 1 treatment group versus vehicle control was performed using one-way ANOVA with Dunnett's multiple comparisons test. Repeat analysis of data was performed independently by distinct investigators to verify treatment effects on a given parameter. See individual figure legends for number of patients and hair follicles analysed per experiment.

## Extended hair follicle organ culture experiments for epithelial bulge stem cell analyses

Hair follicles were obtained from Kosmedklinik Hamburg (Dr. med. Erdmann, Hamburg, Germany) and Provio GmbH, Dr. Gerd Lindner (Dr. rer. nat. Gerd Lindner, Berlin, Germany). Skin samples were obtained after informed consent and ethics committee approvals (University of Muenster) and treated according to the Helsinki Ethical Guidelines for medical research involving human subjects, under human tissue act guidelines. Hair follicles were processed and incubated in hair follicle media as described above then treated with 100 nM paclitaxel (Sigma-Aldrich, T7402) or DMSO (vehicle, 0.3%) at day 2, then again at day 4 after microdissection, before freezing on days 5/6. Cryosections for these experiments were prepared at a thickness of 6 μM.

Primary antibodies: Anti-Cytokeratin 15 (LHK15) (Millipore, #CBL272) (1:200), anti-Ki-67 (Abcam, ab15580) (1:100), Cleaved Caspase-3 (Asp175) (Cell signalling #9661) (1:400), anti-phospho-histone H2A.X (Ser 139) (Cell Signalling, #2577). Secondary antibodies: Goat anti-Rabbit Alexa Fluor 546 (Invitrogen, #A11035), 1:400, goat anti-Mouse Alexa Fluor 488 (Invitrogen, #A11029), 1:400. Tissue was counterstained with nuclear DAPI (4′,6-Diamidine-2′-phenylindole dihydrochloride) (Sigma-Aldrich, #10236276001). Slides and stains were visualised and photo-documented using Keyence BZ-9000E fluorescence microscope. Analysis was performed as above. Extended cultures were performed by Monasterium Laboratory GmbH.

---

### The paper explained

**Problem**

Paclitaxel and docetaxel, taxanes widely used in the treatment of breast, lung, ovarian and cervical cancer, cause chemotherapy-induced alopecia. This is increasingly being seen in the form of permanent chemotherapy-induced alopecia, which poses a tremendous psychosocial burden for cancer survivors. Furthermore, as current treatment options are limited, the urgent need to develop more effective chemotherapy-induced alopecia management strategies remains unmet.

**Results**

In the current study, we have refined a clinically relevant human hair follicle organ culture system, in which we show how taxanes damage stem, progenitor and transit amplifying cell niches in the human hair follicle in a cell cycle-dependent manner. Using this new model system, we probe a novel chemotherapy-induced alopecia management strategy, i.e. hair follicle protection through targeted cell cycle arrest. We show that the G1 arresting CDK4/6 inhibitor palbociclib antagonises the mitosis-targeting cytotoxicity of taxane chemotherapy in stem cell and transit amplifying cell compartments in the human hair follicle, without promoting premature catagen (i.e. hair growth arrest) or additional hair follicle toxicity.

**Impact**

Our newly developed and clinically relevant human model of taxane-induced hair follicle damage provides an invaluable preclinical research tool for developing novel strategies to manage taxane-induced hair loss and protect against human epithelial stem cell toxicity. Furthermore, we provide a proof-of-principle of a novel therapeutic approach, i.e. pharmacological G1 arrest, that could mitigate taxane-induced hair follicle damage and prevent subsequent chemotherapy-induced alopecia, e.g. through the topical delivery of G1 arresting agents.

(S10) (Abcam, ab5176) (1:100), Cleaved Caspase-3 (Asp175) (Cell signalling #9661) (1:50), Anti-Cytokeratin 15 [LHK15] (Abcam, ab80522) (1:500), Anti-Cytokeratin 15 [EPR1614Y] (Abcam, ab52816) (1:500). Goat anti-Mouse/Rabbit Secondary Antibodies, Alexa Fluor 488/594 (Invitrogen, #A11001, #A11005 #A11008 #A11037) (1:200). See also supporting references (Purba et al, 2016, 2017a,b).

## EdU/EU detection

Where indicated, prior to isolation hair follicles were incubated in the presence of 20 μM 5-Ethynyl-2′-deoxyuridine (EdU) for 4 h or 0.5 mM of 5-Ethynyl Uridine (EU) for 2 h during hair follicle organ culture (See Purba et al, 2016, 2017a,b, 2018). Fluorescent detection of incorporated EdU or EU was performed as previously described and as per kit instructions using the Click-iT™ EdU Alexa Fluor™ 594 Imaging Kit (Thermo Fisher, #C10339) or Click-iT™ RNA Alexa Fluor™ 594 Imaging Kit (Thermo Fisher, #C10330), respectively.

## Microscopy and imaging

Slides and stains were visualised and photo-documented using Keyence BZ-8000 fluorescence microscope and software at a

## Study approval

Experiments within this study were performed under ethical approval granted by the University of Manchester (REC reference 19/NW/0082) or University of Muenster (n.2015 -602-f-S).

**Expanded View** for this article is available online.

## Acknowledgements

This work was supported by the NIHR Manchester Biomedical Research Centre ("Inflammatory Hair Diseases Programme") and MRC DTP Research Experience Placement, SEI Learning Through Research and MBChB APEP student programmes at the University of Manchester. Derek Pye is thanked for indispensable technical support. Sandra Rieger is thanked for providing constructive critical feedback on the manuscript.

## Author contributions

TSP: Study design, performance of experiments, and analysis and manuscript preparation. KN, LB, ES and EM: Study design, performance of experiments, and analysis and manuscript editing. NH, AO, CM and JJ: Performance of experiments and analysis and manuscript editing. AS: Provided essential research material. RP: Study design, data interpretation and manuscript editing.

## Conflict of interest

Lars Brunken was supported by a Monasterium Laboratory Master studentship. Ralf Paus is the founder and managing owner of Monasterium Laboratory. The company does not hold any patents in the area covered and has no commercial interest in the compounds used here.

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
