## [Review Process File · EMBO Molecular Medicine]

CDK4/6 inhibition mitigates stem cell damage in a novel model for taxane-induced alopecia

Talveen S. Purba, Kayumba Ng'andu, Lars Brunken Eleanor Smart, Ellen Mitchell, Nashat Hassan, Aaron O'Brien, Charlotte Mellor, Jennifer Jackson, Asim Shahmalak, Ralf Paus

Review timeline:

Submission date:	19 June 2019
Editorial Decision:	19 July 2019
Revision received:	5 August 2019
Editorial Decision:	7 August 2019
Revision received:	13 August 2019
Accepted:	16 August 2019

Editor: Céline Carret

Transaction Report:

1st Editorial Decision

19 July 2019

Thank you for the submission of your manuscript to EMBO Molecular Medicine. I apologise for the delay in reaching a decision. Although I was hoping to obtain a third evaluation, I am now proceeding based on the two consistent evaluations obtained so far as further delays cannot be justified. I will forward Reviewer 2's delayed report, if and as soon as we are able to obtain it. When (within reason) this report does arrive and if it raises additional important issues that have to be addressed to support this study, these would also need to be taken into consideration in your revision. Please note that I would not ask you to consider further-reaching requests with respect to the current evaluations.

As you will see from the comments below, the referees are enthusiastic about the study and have suggestions and recommendations to further improve clarity as well as increase the potential clinical implications with a thorough discussion as suggested, which is particularly important for our scope.

We would therefore welcome the submission of a revised version within three months for further consideration and would like to encourage you to address all the criticisms raised as suggested to improve conclusiveness and clarity. Please note that EMBO Molecular Medicine strongly supports a single round of revision and that, as acceptance or rejection of the manuscript may depend on another round of review, your responses should be as complete as possible.

I look forward to receiving your revised manuscript.

***** Reviewer's comments *****

Referee #3 (Comments on Novelty/Model System for Author):

These are human explants of hair follicles, which are suited for this study

Referee #3 (Remarks for Author):

In this manuscript the authors investigate chemotherapy-induced alopecia by taxanes. This is an interesting topic. The authors convincingly show that taxanes induce a mitotic arrest in hair follicles but overall proliferation is not affected. Eventually, this leads to apoptosis, which is not surprising. Treatment of hair follicles with the CDK4/6 inhibitor palbociclib induced a G1 arrest and protected hair follicles from entering mitosis and apoptosis. The authors tried different regimens with similar results, which underscores that CDK4/6 inhibitor treatment could work in preventing of chemotherapy-induced alopecia maybe even in human patients down the road.

Overall, I enjoyed reading this manuscript since it is present well and contains valuable data. It is an important contribution in the understanding of alopecia and the mechanism of CDK4/6 inhibitors, which are FDA approved to be used in the clinic.

There are a number of issues, which need to be addressed:

1. The authors should try to use less abbreviations...it really makes the reading more difficult than needed and does not serve any real purpose. For example, HF, ORS, CIA (especially this one made me smile!), etc.
2. My feeling is that Figure 1C and 1D should be combined in one graph. It would make reading the data easier.
3. The data for Figure 2D should be quantified as shown in Figure 2A. From the description, the numbers are less impressive than for Paclitaxel but it is still important to have these numbers.
4. For each Figure is should be mentioned in the Figure legends, how many cells were counted. This is essential for the understanding of the data.
5. In Figure 3B, the data is plotted inversely which is confusing. The Paclitaxel treated samples should have a lower count for RNA synthesis.
6. The pictures (data) for Figure 4E needs to be shown. This can be done as supplemental data/figure.
7. The last 3 figures are almost the same but the experiments are different. The authors should try to describe these experiments in a way that keeps the attention of the reader high. Otherwise, the reader gets lost.

Referee #4 (Comments on Novelty/Model System for Author):

Chemotherapy has many devastating side effects on proliferative healthy organs including digestive tract, bone marrow, kidney and skin. Chemotherapy-induced hair loss is one of these highly distressing adverse side effects. The research community is confident that very soon we will significantly increase efficacy of the cancer treatment and in parallel we should address a challenge how to reduce its undesired effects.

In this manuscript, Purba et al. examine the implication of a group of chemotherapeutic agents, Taxanes, in detrimental effects on human scalp hair follicles using ex vivo organ culture model. In addition, for the first time the authors experimentally test G1 cell cycle arrest therapy as a preventive treatment in taxane-induced hair follicle damage.

Presented proof-of-principal experiments are very well organized and very informative together with established methods of detection of proliferation, apoptosis and protein synthesis. Developed ex vivo assay could be very useful for studying and experimentally manipulating toxicology in healthy human hair follicles.

The authors show that mitosis-targeting paclitaxel and docetaxel affect highly proliferative hair forming matrix keratinocytes and Keratin 15 positive epithelial stem/progenitor cell niches located in upper outer root sheath portion of the hair follicle. The authors note, that normally slow cycling stem cells become more proliferative in culture. The fact should be discussed in relevance to

permanent chemotherapy induced hair loss when patient undergoes multiple treatment sessions with possible damage of hair follicle stem cells occurs during activation of a new growth cycle followed by initial chemotherapy-induced dystrophic catagen (Paus et al., Lancet Oncol 2013). Then authors tested the hypothesis that pharmacologically induced cell cycle arrest protects against taxane-induced human hair follicle damage. The G1 arresting CDK4/6 inhibitor, palbociclib, protected paclitaxel-induced cytotoxicity in proliferative epithelial progenitor compartments. It's very exciting model to test further.

1st Revision - authors' response

5 August 2019

***** Reviewer's comments ***** Referee #3.

Thank you so much for your encouraging comments and for providing very helpful constructive suggestions to improve our manuscript.

1. The authors should try to use less abbreviations...it really makes the reading more difficult than needed and does not serve any real purpose. For example, HF, ORS, CIA (especially this one made me smile!), etc.

Done as requested: the abbreviations mentioned (i.e. CIA, ORS, HF) have been removed from the main body of text to enhance readability.

2. My feeling is that Figure 1C and 1D should be combined in one graph. It would make reading the data easier.

Thank you for this suggestion. We have altered the labeling of the graphs to more clearly distinguish data obtained from different experiments to make the data easier to read. Please see revised graphs in Figure 1C and 1D.

As the control data from graphs 1C and 1D are from distinct organ culture experiments (i.e. using hair follicles from unique donors), it would not be ideal to combine this data within the graph as one single control group, as the data would not be directly comparable. However, we'd be happy to generate a combined figure, if the editors advise to do so.

3. The data for Figure 2D should be quantified as shown in Figure 2A. From the description, the numbers are less impressive than for Paclitaxel but it is still important to have these numbers.

Thanks – agreed. As requested, we have performed this quantitative analysis, and are now presenting the data in a new figure (Figure 2B).

4. For each Figure is should be mentioned in the Figure legends, how many cells were counted. This is essential for the understanding of the data.

Within each graph, each value plotted represents the mean number of cells counted for a given parameter (e.g. Ki-67 labeling) per hair follicle analyzed (within a defined reference region, consistently applied to control and treated conditions). This has now been further clarified in the main text figure legends (pages 33-38) and in the methods (page 22).

Including this information (i.e. number of cells counted) again within the legend would be superfluous, unless we have misunderstood the request of the reviewer, in which case we would be happy to accommodate this request upon further clarification.

5. In Figure 3B, the data is plotted inversely which is confusing. The Paclitaxel treated samples should have a lower count for RNA synthesis.

Thanks for pointing this out. We have reanalyzed the data so that the effects of paclitaxel treatment on RNA synthesis are presented more clearly. Please see our new graph in revised Figure 3B.

6. The pictures (data) for Figure 4E needs to be shown. This can be done as supplemental data/figure.

As requested, please see Figure 4E for images corresponding to the quantitative data that is now presented in Figure 4F.

7. The last 3 figures are almost the same but the experiments are different. The authors should try to describe these experiments in a way that keeps the attention of the reader high. Otherwise, the reader gets lost.

Thanks for this suggestion for improvement. We have edited the results section corresponding to the final 3 figures to more clearly describe and distinguish the conditions between the experiments to improve reading clarity. Please see main text, pages 11-15.

Referee #4 (Comments on Novelty/Model System for Author):

Thanks a lot for the very constructive feedback.

The authors show that mitosis-targeting paclitaxel and docetaxel affect highly proliferative hair forming matrix keratinocytes and Keratin 15 positive epithelial stem/progenitor cell niches located in upper outer root sheath portion of the hair follicle. The authors note, that normally slow cycling stem cells become more proliferative in culture. The fact should be discussed in relevance to permanent chemotherapy induced hair loss when patient undergoes multiple treatment sessions with possible damage of hair follicle stem cells occurs during activation of a new growth cycle followed by initial chemotherapy-induced dystrophic catagen (Paus et al., Lancet Oncol 2013).

Thank you for this excellent suggestion. We have now elaborated on this important discussion point within our revised discussion section. Here, we also describe that our *ex vivo* test system is limited in addressing the hair follicle stem cell response to fractionated, repetitive chemotherapy cycles and that mouse models with xenotransplanted human scalp hair follicles that permit long-term observations *in vivo* are required. Please see Discussion, pages 19-20.

Then authors tested the hypothesis that pharmacologically induced cell cycle arrest protects against taxane-induced human hair follicle damage. The G1 arresting CDK4/6 inhibitor, palbociclib, protected paclitaxel-induced cytotoxicity in proliferative epithelial progenitor compartments. It's very exciting model to test further

Thank you again for your positive feedback

2nd Editorial Decision

7 August 2019

Thank you for the submission of your revised manuscript to EMBO Molecular Medicine. I am pleased to inform you that we will accept your manuscript pending editorial amendments.

2nd Revision - authors' response

13 August 2019

The authors performed all minor editorial changes.

Corresponding Author Name: Talveen Purba

Manuscript Number: EMM-2019-11031